# First-Principles Study on the Stability and Electronic Structure of the Charge-Ordered Phase in *α*-(BEDT-TTF)$_2$I$_3$

**Takao Tsumuraya** [1],*[ID]**, Hitoshi Seo** [2,3] **and Tsuyoshi Miyazaki** [4][ID]

1 Priority Organization for Innovation and Excellence, Kumamoto University, 2-39-1 Kurokami, Kumamoto 860-8555, Japan
2 Condensed Matter Theory Laboratory, RIKEN, 2-1 Hirosawa, Wako 351-0198, Japan; seo@riken.jp
3 Center of Emergent Matter Science, RIKEN, 2-1 Hirosawa, Wako 351-0198, Japan
4 International Center for Materials Nanoarchitectonics, National Institute for Materials Science (WPI-MANA) 1-1 Namiki, Tsukuba 305-0044, Japan; miyazaki.tsuyoshi@nims.go.jp
* Correspondence: tsumu@kumamoto-u.ac.jp

**Abstract:** We theoretically study the structural and electronic properties of a molecular conductor, *α*-(BEDT-TTF)$_2$I$_3$, using first-principles density-functional theory calculations, especially in its low-temperature charge-ordered state at ambient pressure. We apply a hybrid functional approach and compare the results with a conventional exchange-correlation functional within the generalized gradient approximation. By performing structural optimization, we found a stable charge-ordered solution for the former, in contrast to the latter approach where the magnitude of the charge imbalance becomes considerably small compared to that when the experimental structure is adopted. The electronic band structure near the Fermi level, with and without structural optimization, as well as the molecule-dependent local density of states of the charge-ordered state are discussed.

**Keywords:** molecular conductor; first-principles calculation; density-functional theory; charge ordering; hybrid functional; electronic structure

## 1. Introduction

Molecular conductors show a rich variety of electronic properties originating from the interplay between the diversity in their crystal structures and the strongly correlated electrons in the valence bands [1,2]. Among them, *α*-(ET)$_2$I$_3$, where ET is the abbreviation for BEDT-TTF = bis(ethylenedithio)tetrathia-fulvalene, is a peculiar example where a (semi)metallic state, in which a massless Dirac-type electronic dispersion is discussed to be realized, competes with a nonmagnetic charge-ordered (CO) insulating state [3,4]. At ambient pressure, the compound undergoes a phase transition by varying the temperature, between the high-temperature metallic and low-temperature insulating state at $T_{CO} = 135$ K [5,6]; the magnetic susceptibility shows a sharp drop below $T_{CO}$, indicating a nonmagnetic ground state [7]. Model calculations predicted a charge ordering owing to Coulomb repulsion between the valence electrons as the origin of this phase transition, with the so-called horizontal CO pattern resulting in a spin-singlet formation between electron spins localized on two molecules in the unit cell [8–12]; such a picture has now been confirmed by different experiments [13–16]. An interesting consequence of the CO is that the pattern breaks the inversion symmetry at high temperatures, and then, the concept called electronic ferroelectricity was developed [17–22].

Although such calculations, using the extended Hubbard model, which is a typical effective model to study strongly correlated electron systems where the basis set is the HOMO of ET molecules, are widely performed, first-principles calculations based on the density-functional theory (DFT) face a difficulty in treating the CO phase. Calculations adopting the crystal structure of the low-temperature CO phase [23,24] succeed in reproducing the insulating gap, as well as the charge disproportionation (CD) among the ET

molecules; however, once structural optimization is performed, the degree of CD becomes considerably small or even vanishes within the numerical accuracy. This is a common problem in molecular conductors showing charge ordering. Recently, we applied a hybrid functional approach using the exchange-correlation functional by Heyd, Scuseria, and Ernzerhof (HSE06) [25–27] to a hydrogen-bonded molecular conductor system $\kappa$-D$_3$(Cat-EDT-TTF/ST)$_2$, where Cat-EDT-TTF is catechol with ethylenedithio-tetrathiafulvalene and Cat-EDT-ST is its selenium-substituted analog [28]. There, the structural stability of the CO insulating phase, which is coupled to the displacements of the deuteriums forming hydrogen bonds, can be reproduced, compared to the widely used DFT method based on the generalized gradient approximation (GGA). This is owed to the more localized nature of the wave functions in the hybrid-functional method; therefore, it is expected that this approach will provide more reliable results than GGA in this class of materials with a pronounced electron-correlation effect.

Here, we apply the HSE06 functional to $\alpha$-(ET)$_2$I$_3$ and compare the results with the conventional exchange-correlation functional by Perdew, Burke, and Ernzerhof (PBE) within GGA, commonly used at present. We investigate the structural stability of the CO phase and, indeed, find a larger degree of CD with the use of HSE06 than GGA-PBE, judging from the central C=C bond of the ET molecules, whose length differences among nonequivalent molecules are closer to, but still smaller than the experimental situation. We further show the calculated electronic structures for the two different functionals and the local density of states (LDOS) probing the charge densities on each ET molecule. The LDOS is obtained as the summation of the density of state (DOS) projected on C and S atoms in each molecule in order to understand the degree of CD between the four molecules in the unit cell.

## 2. Crystal Structure

We refer to the experimental structure at 30 K and 150 K [29], below and above $T_{CO}$, respectively, for which structural optimization for the hydrogen positions was performed. We show the crystal structure at 30 K in Figure 1. Figure 1a shows the view along the $a$-axis where ET layers alternate with layers of iodine ions, I$_3{}^-$. In Figure 1b, a layer of ET molecules in the ac-plane is shown, where the ET molecules form a herringbone pattern consisting of two chains, one with A and A′ molecules stacked along the $a$-axis and the other in which B and C molecules are stacked.

Above $T_{CO}$, the space group is $P\bar{1}$; while the center of mass of the B and C molecules lie on the inversion centers, A and A′ are connected with the inversion operations, e.g., with respect to the inversion center in the middle of A and A′ at (0.5, 0, 0), and these two are crystallographically equivalent. On the other hand, below $T_{CO}$, the system loses the spatial inversion symmetry, and the molecules A and A′ become crystallographically independent, which is considered a consequence of the charge ordering, which is schematically shown in Figure 1b. The structure has a noncentrosymmetric space group of $P1$ [16,29,30].

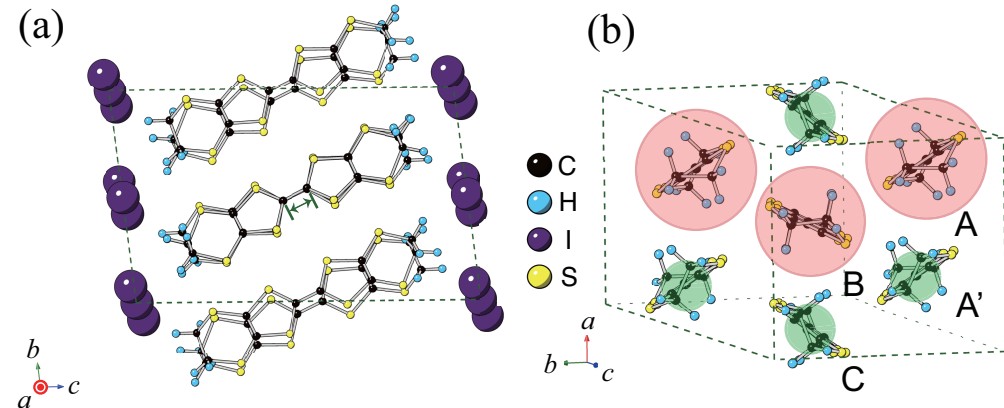

**Figure 1.** Crystal structure of $\alpha$-(BEDT-TTF$_2$)I$_3$ at 30 K [29]. The unit cell is shown by the dashed lines. (**a**) A view along the *a*-axis; (**b**) ET molecules in the *ab*-plane, where four independent molecules, A, A′, B, and C, are shown. The shaded circles schematically show the charge order, whose diameters are drawn to be proportional to the estimated charge values of the molecules in [16]: A = +0.82, A′ = +0.29, B = +0.73, and C = +0.26.

## 3. Calculation Methods

In the present DFT calculations, the Kohn–Sham equations were solved using the pseudopotential method based on the projected augmented plane wave (PAW) method with plane wave basis sets implemented in the Vienna Ab initio Simulation Package [31,32]. In the HSE06 calculations, we first obtained a converged charge density from the self-consistent calculations within GGA, and then, using the GGA charge density as the initial state, the self-consistent hybrid functional calculations were performed. The common *k*-point sampling, including that for the structural optimization, was set as $4 \times 4 \times 2$. The cutoff energy for plane waves was 29.4 Ry for the HSE06 calculations. The range-separation parameter in the HSE06 calculations was 0.2 Å$^{-1}$, and 25% of the exact exchange was mixed to the GGA exchange for the short-range interactions. We set the *k*-point mesh for calculating DOS as $8 \times 8 \times 2$ in both the HSE06 and GGA-PBE calculations.

## 4. Structural Properties

First, we discuss the structural properties by optimizing the internal coordinates using the GGA-PBE and HSE06 functionals. Table 1 lists the optimized bond parameters of the central C=C bond distance of the ET molecules, which reflects the charge density on them [16], together with the experimental values. For reference, we also list the valence of the molecules estimated by an empirical method used in the literature [16,29,33].

For the centrosymmetric ($P\bar{1}$) case above $T_{CO}$, one can see two points compared to the experiments: the average value in GGA-PBE gives a closer value than HSE06. We note that the experimental values are for a high temperature where thermal vibration should be significant, whereas the DFT calculations are basically for the ground state; we should be cautious about a direct comparison. More importantly, HSE06 captures the feature in the experimental values of the C=C bond lengths that shows A $\simeq$ B > C, reflecting the hole density, i.e., the amount of charge on each molecule, in clear contrast to the GGA-PBE case showing A > B $\simeq$ C. We note that the charge densities using the room temperature structure were evaluated [23,34] by Mulliken charge analyses, which agreed with the experimental estimations.

Now, let us discuss the results for the low-temperature CO phase. In the experiments, owing to the charge ordering, CD takes place among the two equivalent A molecules and results in two distinct A and A′ with a difference in the C=C bond lengths of about 0.02–0.03 Å. As mentioned in the Introduction, the present results with GGA-PBE indeed show almost the same values for the A and A′ molecules. In clear contrast, when we use the HSE06 hybrid functional, the difference in the C=C bond lengths between the A and A′

molecules is about 0.005 Å. Moreover, similar to the high-temperature phase, HSE06 gives a shorter C=C bond length for the C molecule than that for the B molecule; in GGA-PBE, they become almost the same. These HSE06 results are consistent with the CO pattern suggested in previous works, theoretically and experimentally, where the charge rich (poor) sites are A and B (A′ and C) with longer (shorter) C=C bond lengths [16,29,35]. We should note that discrepancies still exist from the experimental values, as seen in Table 1: longer C=C bond lengths of C molecules in general seen in the valence as well, a small change in that of B molecule across the phase transition, etc. However, we consider that the HSE06 functional provides the structural stability of the CO state in $\alpha$-(ET)$_2$I$_3$, i.e., a reasonable difference in the A and A′ molecules upon structural optimization.

**Table 1.** Experimental and theoretically-optimized structural parameters for the central C=C double bonds ($d$) and the valence estimated by an empirical method ($Q$) [33] in ET molecules, above (top rows) and below (bottom rows) $T_{CO}$.

| Method | A $d$ (Å) | A′ $d$ (Å) | B $d$ (Å) | C $d$ (Å) | A $Q$ | A′ $Q$ | B $Q$ | C $Q$ |
|---|---|---|---|---|---|---|---|---|
| GGA-PBE | 1.376 | (= A) | 1.373 | 1.373 | 0.75 | (= A) | 0.75 | 0.71 |
| HSE06 | 1.366 | (= A) | 1.364 | 1.360 | 0.80 | (= A) | 0.83 | 0.71 |
| Exp. (150 K) [29] | 1.375 | (= A) | 1.376 | 1.368 | 0.71 | (= A) | 0.74 | 0.48 |
| GGA-PBE | 1.375 | 1.374 | 1.373 | 1.374 | 0.79 | 0.74 | 0.76 | 0.68 |
| HSE06 | 1.368 | 1.363 | 1.364 | 1.361 | 0.94 | 0.79 | 0.84 | 0.73 |
| Exp. (20 K) [16] | 1.384 | 1.361 | 1.385 | 1.360 | 1.00 | 0.43 | 0.86 | 0.32 |
| Exp. (30 K) [29] | 1.382 | 1.363 | 1.381 | 1.361 | 0.90 | 0.45 | 0.89 | 0.39 |

## 5. Electronic Structure

Next, we discuss the electronic states, for the experimental structures above and below $T_{CO}$ and the optimized structures, comparing the two functionals. Figure 2a,b shows the band structure and the total DOS calculated with HSE06 and GGA-PBE, respectively, using the experimental structure above $T_{CO}$, at 150 K. The main features are consistent with the previously reported results, including the extended Hückel approach [36]: the four bands close to the Fermi level are basically attributed to the HOMO of the four constituent ET molecules in the unit cell [23,34,37,38]. Although it is not the focus of this study, we note that there is no consensus in the detailed band structure of this material in the high-temperature phase even within GGA-PBE. Some show a semimetallic state [23,34,38] as in the extended Hückel results, while others suggest a massless Dirac electron-type dispersion with the Fermi level at the Dirac point [29,37]; our results are consistent with the latter.

Comparing the band structures obtained by HSE06 and GGA-PBE, we found that the HSE06 band structure had a wider bandwidth, of about 20%, in spite of the fact that the atomic positions used in the calculations were the same. This trend was in agreement with our results for a proton-coupled system in $\kappa$-D$_3$(Cat-EDT-TTF/ST)$_2$ [28]. As for the case of the optimized atomic coordinates, the results for the HSE06 and GGA-PBE functionals are plotted in Figure 2c,d, respectively. For both functionals, structural optimization does not change the band structures much; the essential features in the band dispersions and the DOS were maintained. The tendency that the HSE06 results give a wider bandwidth than the GGA-PBE was also the same as in the experimental structure.

We then show the band structure and DOS in the low-temperature phase below $T_{CO}$ using the experimental structure, with the HSE06 and PBE-GGA functionals, in Figure 3a,b, respectively. Compared to the result above, the size of the band gap increase with both functionals. The size of the band gap below $T_{CO}$ with HSE06, about 0.06 eV, is larger than the value by GGA-PBE, about 0.04 eV. Similar to the case above $T_{CO}$, the bandwidth of the HSE06 band structure, evaluated from the DOS, is wider than that of GGA-PBE. Despite these quantitative differences in the band gaps and the bandwidths, GGA-PBE

and HSE06 provide similar band structures. Figure 3c,d shows the HSE06 and GGA-PBE band structures for the optimized structure. By structural optimization, the band gap is reduced from the results using the experimental structure: in the HSE06 case, 0.06 eV→0.05 eV, and in GGA-PBE, 0.04 eV→0.02 eV. The smallness of the band gap in GGA-PBE is consistent with the structural properties discussed in the previous section, i.e., the small difference in the central C=C bond lengths among the ET molecules. We note the experimental estimates [20]: from optical measurements, the gap is estimated as 0.075 eV without noticeable in-plane directional dependence, while the DC electric conductivity gave 0.04 eV and 0.08 eV depending on the field direction.

Figure 4 shows the LDOS below $T_{CO}$, which explicitly indicates the CD among the molecules in the unit cell. As mentioned above, we obtained the LDOS as a summation of the projected DOS on the orbitals of C and S atoms in each ET molecule. Figure 4a,b shows the results for HSE06 without and with structural optimization, respectively, and Figure 4c,d shows the same for GGA-PBE. The CD is seen by the larger LDOS in the top band for A and B molecules than A′ and C molecules and vice versa for the second band from the top. Since the Fermi level resides between the two bands, these features directly indicate the larger (smaller) electron density in the A′ and C (A and B) molecules: the CO state is formed by the horizontal stripes along the hole-rich A and B sites. We show the estimations of the valence of each molecule in Table 2, based on the integration of the LDOS. Note that these values are different from $Q$ in Table 1. The GGA-PBE values for the experimental structure were in good agreement with a previous evaluation using a different DFT approach [23]. Comparing the HSE06 LDOS without and with optimization, the CD between A and A′ molecules remains after structural optimization. On the other hand, in the case of GGA-PBE, the CD between A and A′ is smaller after the optimization, where the LDOS, especially in the unoccupied state, is almost the same after the structural optimization (Figure 4d).

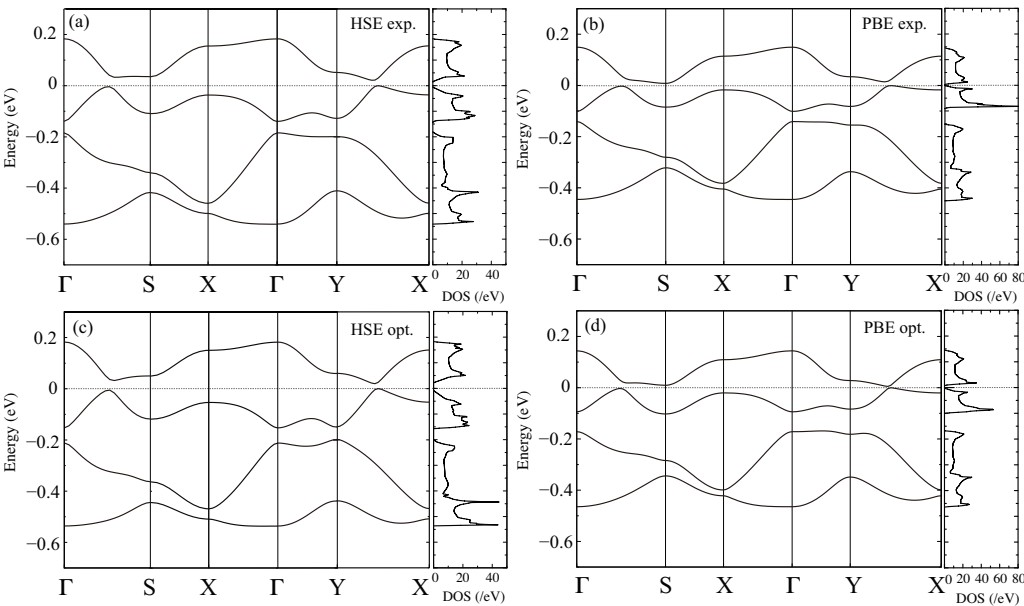

**Figure 2.** The band structure and the total DOS for the experimental structure at 150 K above the CO phase transition calculated with the (**a**) HSE06 and (**b**) GGA-PBE functionals and those for the geometrically optimized structure with the (**c**) HSE06 and (**d**) GGA-PBE functionals.

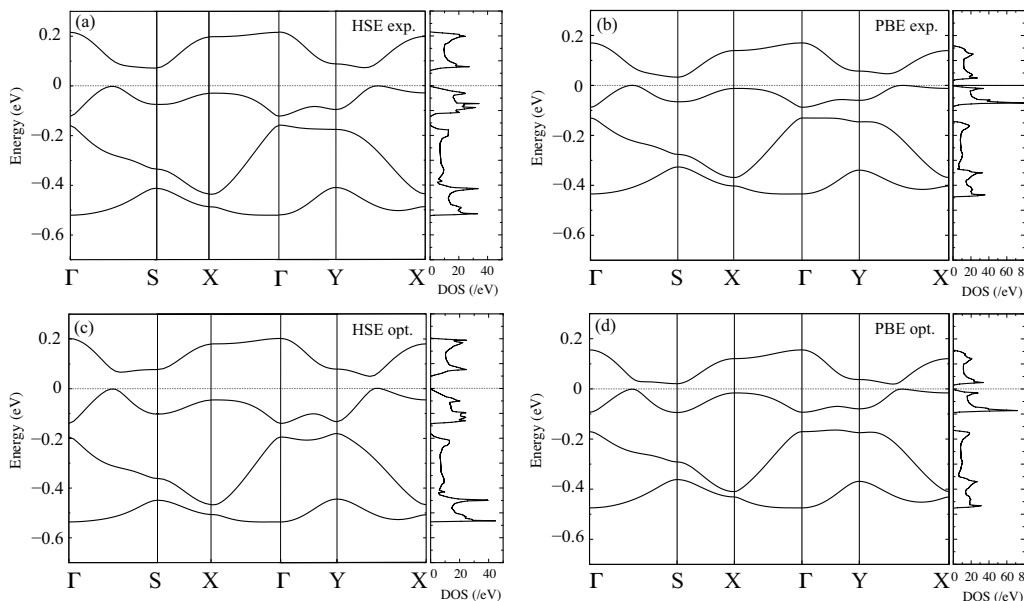

**Figure 3.** The band structure and the total DOS for the experimental structure at 30 K, in the CO phase calculated with the (**a**) HSE06 and (**b**) GGA-PBE functionals and those for the geometrically optimized structure with the (**c**) HSE06 and (**d**) GGA-PBE functionals.

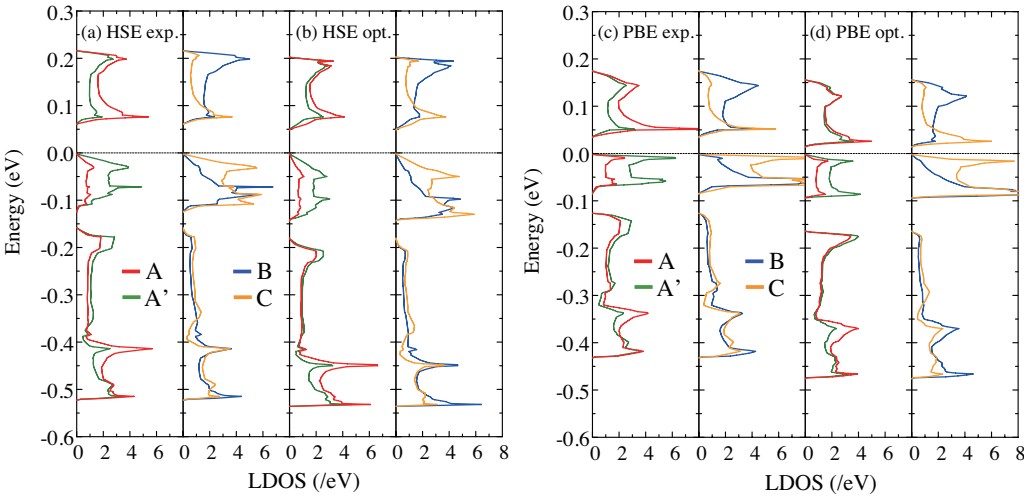

**Figure 4.** The LDOS in the low-temperature CO phase for the four molecules in the unit cell A, A′, B, and C. (**a**) The LDOS calculated with HSE06 for the experimental structure and (**b**) for the geometrically optimized structure. (**c**) The LDOS calculated with GGA-PBE for the experimental structure and (**d**) for the geometrically optimized structure. The plotted LDOS contains four parts: the red, green, blue, and orange curves indicate that of A, A′, B, and C, respectively. The dotted lines at 0 eV show the top of the valence bands.

**Table 2.** The valence of the four molecules in the unit cell, i.e., the hole density obtained by integrating the LDOS shown in Figure 4.

| Method | Structure | A | A′ | B | C |
|---|---|---|---|---|---|
| GGA-PBE | Exp. | 0.72 | 0.42 | 0.64 | 0.32 |
| | Opt. | 0.58 | 0.51 | 0.58 | 0.42 |
| HSE06 | Exp. | 0.62 | 0.40 | 0.68 | 0.32 |
| | Opt. | 0.64 | 0.50 | 0.58 | 0.38 |

## 6. Conclusions

We studied the structural and electronic properties of the molecular conductor $\alpha$-$(ET)_2I_3$ using GGA-PBE and a range-separated hybrid functional of HSE06 and discussed the electronic structure above and below $T_{CO}$. Below $T_{CO}$, HSE06 captures the experimental feature of different molecules (A and A′) showing different C=C bond lengths, reflecting the charge density on each molecule, compared to the GGA-PBE case, where the charge ordering below $T_{CO}$ is obscured when structural optimization is performed. The electronic structure calculations are consistent with such results, showing clear charge disproportionation between A and A′ in the case of HSE06 after structural optimization.

**Author Contributions:** T.T. performed all the calculations and data analyses. All the authors (T.T., H.S. and T.M.) equally contributed to the planning, discussion, and writing of the manuscript. All authors have read and agreed to the published version of the manuscript.

**Funding:** This research was funded by a Grant-in-Aid for Scientific Research (JP16K17756, JP19K04988, JP19K21860, JP19K03723, JP20H00121, JP20H04463) from the Japan Society for the Promotion of Science (JSPS) and JST, CREST Grant Number JPMJCR2094, Japan. This work was performed under the GIMRT Program of the Institute for Materials Research (IMR), Tohoku University (Proposal No. 18K0090). T.T. was supported in part by the Leading Initiative for Excellent Young Researchers (LEADER), a program of the Ministry of Education, Culture, Sports, Science and Technology, Japan (MEXT).

**Data Availability Statement:** Not applicable.

**Acknowledgments:** The study's computations were mainly conducted using the computer facilities of ITO at Kyushu University, MASAMUNE at IMR, Tohoku University, and ISSP, University of Tokyo, Japan.

**Conflicts of Interest:** The authors declare no conflict of interest.

**Sample Availability:** The experimental structures used in this study are available from the CCDC database, where the deposition numbers are 2008980 and 2008981.

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
