# Peer review of "First-Principles Study on the Stability and Electronic Structure of the Charge-Ordered Phase in α-(BEDT-TTF)2I3"

_crystals, doi:10.3390/cryst11091109_

Round 1
Reviewer 1 Report
In this manuscript the authors present the results of density functional (DFT) calculations for alpha-(ET)2I3, which has a charge ordered low temperature phase. They compare the results of a conventional exchange-correlation functional with a more accurate hybrid functional approach. They claim that the hybrid functional is able to capture the low temperature ordered state in this material which is missed by the conventional exchange-correlation method.
Using DFT to understand the electronic properties of strongly-correlated materials is difficult, and I think that these results provide a useful comparison on a relatively well-characterized material and should therefore be published in Crystals. I do however think that the manuscript could be improved before publication:
1. It would be very helpful to indicate on Fig 1(b) the relative charge density on the different molecules in the ordered phase. Different color shading, superimposed circles showing relative charge density, or some other technique could be used.
2. In the abstract they claim that the conventional exchange-correlation DFT gives a "magnitude of charge imbalance" that is "considerably small compared to that when the experimental structure is adopted." They further note on page 3 line 94, that the charge densities they calculate are consistent. However, no results of the charge density analysis are reported at all. These results should be reported for both the hybrid and conventional DFT.
3. It is not clear from the data presented that the hybrid DFT accurately captures all features of the low temperature state. For example, there is no change in the C=C bond length of molecule B within HSE06 between high and low temperatures, while there is a large change experimentally. The authors should comment.
Reviewer 2 Report
The manuscript of Tsumuraya and coworkers describes theoretical first-principles calculations performed on a molecular conductor based on the BEDT-TTF organic donor. The aim of this work is to compare the experimental data with calculations using two different approaches. The new approach proposed by the authors explains some experimental features that are not reproduced by the conventional methods. For this reason I find this contribution interesting for the community of Crystals and I recommend publication after minor revisions.
- When mentioning the system D3(Cat-EDT-TTF/ST)2, the full name of the involved building blocks should be reported for those readers that are not familiar with these molecules.
- The meaning of LDOS and DOS are taken fro granted but should be defined when cited for the first time in the text.
- In the conclusions, the authors says "capture the features in ...". This sounds too vague to my and should be rephrased with a more explicit formulation as done in the description of the results.
- The conclusions are extremely short and do not address any comment on the considerations made in the "Electronic Structure" session, while concerning only the "Structural Properties". The author should add a brief comment on that to summarize the findings and provide an overall summary of the work.
Round 2
Reviewer 1 Report
In the revised manuscript the authors have satisfactorly addressed all of my previous comments.
This manuscript is a resubmission of an earlier submission. The following is a list of the peer review reports and author responses from that submission.
Round 1
Reviewer 1 Report
This manuscript reports a DFT study of the electronic structure of a-(BEDT-TTF)2I3. This material has been the subject of several DFT studies in the past using the classical PBE functional. The authors have carried out structural optimizations with a different functional, the hybrid HSE06 one, and conclude that the hybrid functional provides a better description of the low temperature charge ordered state than the classical PBE. Unfortunately, this referee does not find any reason in the manuscript to support this claim.
1) To begin with, the calculations with the new functional lead to a fatal error when considering the system before the transition. The system is a semimetal, something that all previous DFT calculations described correctly (present references 23, 24 and others) whereas the new functional does not. Either using the optimized or experimental structure, the HSE06 functional leads to a semiconducting gap (according to Figure 2) above the transition. All previous calculations using the experimental structure and the classical PBE functional lead to a semimetallic state. Then, if the HSE06 functional is not even able to describe such a fundamental property of the material, how can we accept that it is more appropriate?
When refereeing to these calculations the authors state (line 113): “The main features are consistent with the previous reported results”. This is FALSE, they do not reproduce the more fundamental aspect of these calculations: the change in conductivity regime across the transition (which by the way, in the previous calculations it is in agreement with the experimental situation).
2) When discussing the results of Table 1 the authors report and state in the paper that for the average structure the classical PBE calculations give a closer agreement with the experimental results (then they try to dismiss this finding with a not very sound comment about vibrations, why?) but since PBE calculations give not only a good structural description, as stated by the authors, but ALSO the correct answer with respect to the very fundamental observation of a semimetallic vs semiconducting behavior across the transition, any responsible scientist should conclude that, at least above and across the transition, the PBE functional gives a better description of the material.)
The authors say that the HSE06 captures better the experimental situation because the C=C bondlengths of the A, B and C molecules reflects better the charge of the molecule A≅B>C in contrast with the GGA result A>B≅C. I do not understand why they make this comment. First, the C=C alone does not explain the “charge” of BEDT-TTF donors. As it has been known for a longtime it is a combination of the two types of C=C bonds and the two types of C-S bonds that must be taken into account to explain the charges (Guionneau-Day equation). Second, I have looked at the previous PBE calculations just to be sure of what are the results and, for instance, I found in reference 23 that the calculated density of holes is +0.52 (A), +0.54 (B) and +0.32 (C). These values are in excellent agreement with what they quote as the experimental trend. The claim of the authors is not at all justified, in fact is incorrect. Again, my conclusion is that the classical PBE calculations give a nice description of the system, in contrat with the statements of the authors.
Finally, when the authors consider the structure below the transition, when looking at the lower part of Table 1 one can find that the C=C distances are better for two of the donors but the PBE are better for the other two. How the authors can conclude that the HSE06 results are better?
Of course, this referee is not interested in supporting one or the other DFT functionals but he firmly believes that what is reported in a scientific publication must be justified by clear data and fair comparison with already existing works.
3) Why the authors insist in the importance of the optimized structure? The experimental structures above and below the transition are known, why they need to optimize the structure? Why they need a less accurate structure? For formal reasons, I could understand this if they were going to do a phonons study but not for what they are discussing here. It is a well-known fact that the structure of these materials is strongly dependent of how the non-bonded interactions are taken into account and there is no consensus on what functional is better for a given type of system when nonbonded interactions can be important. In this referee experience the HSE06 functional is usually not the better for structural optimizations. This functional was made to produce good gaps in semiconductors. There is no reason to assume that will give better crystal structures. In addition, the authors should have indicated how did they carry out the optimizations: did they optimize also the cell constants or not?
4) A great deal of information is available concerning the CO state for this system. For instance, the optical and conductivity gaps for different directions are known (Dressel and coworkers), etc. Why they do not compare the present results with the experimental information?
5) What is the physical origin of the charge separation? In references 23 and 24 I found a great deal of understanding about the physics of the system but nothing here.
In summary, the main claims of this manuscript are not justified according to the data reported, the functional used cannot even account for the change in conductivity regime of the system, and no physical insight is gained from the present calculations. It seems to this referee that already existing papers in the literature already give a more balanced, insightful and factually correct description than in the present work. Under such circumstances, the present manuscript cannot be recommended for publication.
Reviewer 2 Report
The presented manuscript reports band structure of a molecular conductor.
In general the manuscript seems to be a functional benchmark data with poor justified chemical background. I'm not sure such paper could be in scope of interest for the readers of the journal.